# Mesenchymal Stem Cell Microvesicles from Adipose Tissue: Unraveling Their Impact on Primary Ovarian Cancer Cells and Their Therapeutic Opportunities

**DOI:** 10.3390/ijms242115862

**Published:** 2023-11-01

**Authors:** Agnieszka Szyposzynska, Aleksandra Bielawska-Pohl, Marek Murawski, Rafal Sozanski, Grzegorz Chodaczek, Aleksandra Klimczak

**Affiliations:** 1Laboratory of Biology of Stem and Neoplastic Cells, Hirszfeld Institute of Immunology and Experimental Therapy, Polish Academy of Sciences, 53-114 Wroclaw, Poland; agnieszka.szyposzynska@hirszfeld.pl (A.S.); aleksandra.bielawska-pohl@hirszfeld.pl (A.B.-P.); 21st Department of Gynecology and Obstetrics, Wroclaw Medical University, 50-599 Wroclaw, Poland; marek.murawski@umw.edu.pl (M.M.); rafal.sozanski@umw.edu.pl (R.S.); 3Bioimaging Laboratory, Łukasiewicz Research Network-PORT Polish Center for Technology Development, 54-066 Wroclaw, Poland; grzegorz.chodaczek@port.lukasiewicz.gov.pl

**Keywords:** ovarian cancer, primary ovarian cancer cells, mesenchymal stem cells, extracellular vesicles, MSC-microvesicles

## Abstract

Mesenchymal stem cells (MSCs) and their derivatives can be promising tools in oncology including ovarian cancer treatment. This study aimed to determine the effect of HATMSC2-MVs (microvesicles derived from human immortalized mesenchymal stem cells of adipose tissue origin) on the fate and behavior of primary ovarian cancer cells. Human primary ovarian cancer (OvCa) cells were isolated from two sources: post-operative tissue of ovarian cancer and ascitic fluid. The phenotype of cells was characterized using flow cytometry, real-time RT-PCR, and immunofluorescence staining. The effect of HATMSC2-MVs on the biological activity of primary cells was analyzed in 2D (proliferation, migration, and cell survival) and 3D (cell survival) models. We demonstrated that HATMSC2-MVs internalized into primary ovarian cancer cells decrease the metabolic activity and induce the cancer cell death and are leading to decreased migratory activity of tumor cells. The results suggests that the anti-cancer effect of HATMSC2-MVs, with high probability, is contributed by the delivery of molecules that induce cell cycle arrest and apoptosis (p21, tumor suppressor p53, executor caspase 3) and proapoptotic regulators (bad, BIM, Fas, FasL, p27, TRAIL-R1, TRAIL-R2), and their presence has been confirmed by apoptotic protein antibody array. In this study, we demonstrate the ability to inhibit primary OvCa cells growth and apoptosis induction after exposure of OvCa cells on HATMSC2-MVs treatment; however, further studies are needed to clarify their anticancer activities.

## 1. Introduction

Ovarian cancer (OvCa) is the global seventh most common cancer in women and eighth most common cause of death from cancer in women worldwide (International Agency for Research on Cancer, World Health Organization [1]. OvCa includes a heterogeneous group of malignancies that differ in etiology, cell of origin, phenotype, pathological grade, risk factors, prognosis, response to therapy, and numerous other characteristics [2]. Depending on the cellular origin, OvCa are divided into epithelial and non-epithelial subtypes. Epithelial cancers are most common, account for about 90% of OvCa, and are classified by tumor cell histology as: serous (52%), endometroid (10%), mucinous (6%), clear cell carcinoma (6%), and other miscellaneous and more rare subtypes [2]. Based on clinicopathologic features, epithelial carcinomas are classified into two subtypes: Type I, unilateral, considered as low-grade with good prognosis of survival, and Type II representing high-grade malignancies, characterized by involvement of both ovaries, aggressive behavior, and low rate of survival. The most often diagnosed type of epithelial OvCa is ovarian serous carcinoma; 90% of serous subtypes are classified as high grade, and only 10% are present as low grade. Another type of OvCa is ovarian mucinous carcinoma which consists of benign and malignant cells, and clear cell carcinoma with low proliferative activity [3]. A relatively rare cancer is ovarian cystadenofibroma, which contains two types of cancer cells: epithelial and fibroblast stromal cells [3].

About 5% of patients with OvCa die, and the mortality is associated with advanced stage of tumor due to the postponed examination of diseases and inaccessibility of screening tests. Moreover, at the beginning of the disease’s development, none of the signs are observed and the patients are diagnosed at the advanced stages usually with metastasis to distant organs. The choice of OvCa treatment method depends on several factors including classification of cancer, current stage, and the patient’s condition and genetic background. The widely used method is chemotherapy followed by surgery or neoadjuvant therapy after surgery [4]. Immunotherapy using a combination of anti-angiogenic drugs, inhibitors of signaling pathways, or vaccines with modified T cells are used in clinical trials of OvCa; however, these approaches are still under experimental procedures [3,5].

Current conventional cancer therapies target cancer cells that are abnormally proliferating; however, these strategies are not fully effective due to the presence of cancer stem cells (CSCs) showing specific biological behavior including high resistance to radio-/chemotherapy and slower proliferation rate that leads to the treatment failure. CSCs represent a rare population of cells in tumors, typically less than one percent. CSCs have similar properties to normal stem cells such as self-renewal, the potential for differentiation, slow proliferation, and activity of aldehyde dehydrogenase 1 (ALDH1) [6]. Moreover, CSCs have features that make them aggressive and resistant to widely used drugs in cancer treatment—especially the ability to migrate from tumor stroma to surrounding tissues, escape from the activity of the immune system, and resistance to pro-apoptotic signals. OvCa stem cells are characterized by the presence of characteristic markers CD133, CD44, CD117 (c-kit), and the activity of ALDH1 [4]. CD133 (Prominin-1) is a glycoprotein involved in different processes such as cellular metabolism, autophagy, and apoptosis. CD133-positive cells in the tumor niche are chemo- and radio-resistant and have higher tumorigenic potential and potential to metastasize compared to cancer cells [7]. Overexpression of CD44 (hyaluronic acid-binding glycoprotein) is observed in different types of tumors [8]. CD117 is a tyrosine kinase receptor that plays a crucial role in signaling pathways responsible for cell proliferation, survival, and migration [9]. ALDH1 is responsible for the oxidation of intracellular aldehydes and directly related to the survival and chemoresistance of CSCs [10,11].

Cancer cells in tumor parenchyma do not exist as self-sustained entities and need for survival a non-neoplastic microenvironment, comprised of a heterogeneous network of different types of cellular and extracellular components including: tumor vasculature, inflammatory cells, tumor-associated fibroblasts, extracellular matrix, and mesenchymal stem/stromal cells [12]. In tumor niches, the most important types of cell-to-cell communication are direct contact between cells, presence of gap junctions, secretion of soluble factors, and release of extracellular vesicles (EVs). Thus, inhibition of the biological activity of cancer cells, including CSCs, would be a promising and beneficial approach in cancer therapy, and one of them is the idea of using cytotherapy with native or modified mesenchymal stem/stromal cells (MSCs) [12].

Mesenchymal stem/stromal cells (MSCs) represent a heterogeneous population of multipotent cells that are present in different adult tissues and contribute to organ homeostasis. Isolated MSCs play an important role in regenerative medicine due to their increased proliferative activity and survival, anti-inflammatory potential, and modulation of the immune system [13]. Moreover, MSCs also release various bioactive molecules in the form of soluble factors, EVs, exosomes, and microvesicles (MVs), which act in the tissue microenvironment as mediators of cell-to-cell communication by exerting paracrine effects [14,15]. It is well documented that MSCs are recruited to the tumor site, and the tropism of MSCs to tumors has been used as the basis of cancer cytotherapy [12].

Different research reported conflicted results related to the anti-tumorigenic or pro-tumorigenic activity of MSCs and their derivatives in OvCa in vitro and in vivo [16,17,18]. The co-injection of umbilical cord-derived MSC (UC-MSC) with SKOV-3 (human ovarian adenocarcinoma) cells into NOD/SCID mice leads to an increase in tumor growth compared to SKOV-3 cells injections alone [18]. On the other hand, UC-MSC leads to a reduction in the survival rate of co-cultured ovarian carcinoma cells (CAOV3) [17]. Anti-proliferative and anti-tumor activity of MVs produced by bone marrow-derived MSCs (BM-MSCs) on SKOV3 cell line and in the SCID mice model was reported [16]. It was documented that the biological/therapeutic effect of MSCs in cancer therapy depends on the tissue origin of MSCs, the application of native or genetically modified MSCs, and also on the histological type of OvCa [19,20].

Importantly, MSCs are used in clinical trials not only for OvCa treatment but also in other types of solid tumors. In the completed first stage of the clinical trial (NCT02530047), MSCs isolated from healthy male donors overexpressing interferon β were used for the treatment of advanced epithelial OvCa. The administration of MSCs once a week was performed by intraperitoneal infusion. However, the results are still being analyzed and have not yet been published.

Our recent studies showed that MVs isolated from MSCs of human adipose-tissue origin (HATMSC-MVs) induce apoptosis in the OvCa cell lines OAW-42 and ES-2 [21]. Based on our experience that the transfer of HATMSC-MVs into OvCa cell lines results in inhibition of cells proliferation and apoptosis induction [21], we decided to explore this biological phenomenon on post-operative primary OvCa cells. This study aims to determine the effect of HATMSC-MVs on the biological activity of primary ovarian tumor cells, derived from post-operative tissue of OvCa (post-op. of OvCa) and ascitic fluid, in 2D and 3D models.

## 2. Results

### 2.1. Characteristics of Primary Ovarian Cancer Cells

Primary OvCa cells isolated from the post-op. tumor and/or ascitic fluid of 16 patients were characterized for MSCs phenotype (CD73, CD90, CD105), CSCs markers (CD24, CD44, CD133), and hematopoietic stem and progenitor marker (CD34) and leukocyte common antigen (CD45). Evaluation of pluripotency-related markers (Oct4, Sox2, Nanog) and proto-oncogenic transcripts (p53, p21, c-myc) has been performed to assess biological features of examined OvCa cells.

#### 2.1.1. Phenotype of Primary Ovarian Cancer Cells

Flow cytometry analysis for the expression of MSCs markers for OvCa cells of post-op. high-grade serous OvCa (N = 9) revealed that CD73 expression is present on a subpopulation of cells ranging from 2.0% to 90.0% with a median value 82.0%. Similarly, CD90 was detected on cancer cells from 9.0% to 100.0% of the population with a median value of 80.0%, whereas CD105 was present from 4.0% to 85.0% of cancer cells population with a median value of 32.0%. Cancer stem cells markers represented by CD133 expression were present in a broad range from 0.0% to 46.0% with a median value of 0.5%; CD44 from 20.0% to 100.0% (median value 91.0%); and cells with CD24 expression were present from 0.0% to 16.0% (median value 7.0%). Cells expressing hematopoietic stem and progenitor marker CD34 (median value 0.5%) and leukocyte common antigen CD45 (median value 10.0%) were represented occasionally (Figure 1a). The median value of the subpopulation of cells expressing MSCs markers, isolated from corresponding to cancer tissue ascitic fluid (N = 9), was assessed for: CD73 (50.0%), CD90 (78.0%), and CD105 (22.0%). The fraction of cells with CSCs phenotype was represented by CD133 (median 2.0%), CD44 (median 78.0%), and CD24 (median 3.0%) (Figure 1a).

The analysis of membrane markers performed for cells derived from other histological types of OvCa (N = 4) revealed the subpopulation of CD73-positive cells ranging from 34.0% and 93.0% with a median value of 57.5%. CD105 was detected in a broad range of cells from 7.0% to 85.0% with a median value of 22.5%. The cells isolated from post-op. tissues of other histological types of OvCa also expressed CSCs such as CD133, CD44, and CD24. CD133 expression was present in a subpopulation of cancer cells from 1.1% to 18.0% with a median value of 6.5%; CD44 from 36.0% to 97.0% (median value 80.0%); and cells with CD24 expression consist of 0.0% to 16.0% (median value 7.0%) of the whole population of isolated cancer cells. Moreover, a small fraction of cells was positive for hematopoietic marker CD45 (median value 6.5%), and all isolated cells were negative for CD34.

For cells derived from corresponding to cancer tissue ascitic fluid (N = 3), the positive population of cells with MSCs phenotype CD73, CD90, and CD105 was detected. The median values of positive cancer cells for CD73, CD90, and CD105 were 47.0%, 71.0%, and 25.0% respectively. Importantly, the positive populations for CSCs markers such as CD133, CD44, and CD24 were determined in the whole population of cancer cells. The highest median value was observed for the CD44 marker (96.0%). The median for CD133 and CD24 was up to 10.0% (2.0% and 10.0%, respectively). Moreover, isolated cells were negative for hematopoietic marker CD34; however, a fraction of cells was positive for CD45 (median value 22.0%) (Figure 1a).

Flow cytometry analysis of high-grade serous OvCa and cystadenofibroma, classified in this study to other histological types of OvCa, are illustrated on representative histograms (Figure 1b).

#### 2.1.2. mRNA Expression for Pluripotency-Related, Protooncogenic, and CSCs Markers

The real-time RT-PCR was performed for the expression of pluripotency-related transcripts (*Oct4*, *Sox2*), protooncogenic transcripts (wild-type *p53*, *p21*, *c-myc*), and *CD133* marker on cells derived from 14 patients (Figure 2 and Figure 3).

The mRNA gene expression for pluripotency markers *Oct4* and *Sox2* in cells was detected (median of 2^−ΔCT^ was 3.35 × 10^−5^, and 6.66 × 10^−6^, respectively). Moreover, the expression of protooncogenic transcripts wild-type *p53*, *p21*, and *c-myc* was present (median of 2^−ΔCT^ was 5.64 × 10^−4^, 2.18 × 10^−3^, and 3.51 × 10^−3^, respectively). For cells from ascitic fluid of high-grade serous OvCa (N = 8), the expression of *Oct4* and *Sox2* was also observed (median of 2^−ΔCT^ was 5.47 × 10^−5^ and 1.07 × 10^−5^, respectively). The level of protooncogenic markers *p53*, *p21*, and *c-myc* was determined (median of 2^−ΔCT^ was 1.03 × 10^−3^, 3.13 × 10^−3^, and 4.94 × 10^−3^, respectively) (Figure 2).

The same analysis was performed for cells derived from other histological types of OvCa. For cells derived from post-op. tumor of OvCa (N = 5) the expression of pluripotency-related markers *Oct4* and *Sox2* was detected (median of 2^−ΔCT^ was 5.38 × 10^−5^ and 1.22 × 10^−5^, respectively). Also, the proto-oncogenic transcripts *p53*, *p21*, and *c-myc* were determined (median of 2^−ΔCT^ was 7.57 × 10^−4^, 1.86 × 10^−3^, and 2.62 × 10^−3^, respectively). For cells from ascitic fluid (N = 2), the expression of *Oct4* and *Sox2* was also present (median of 2^−ΔCT^ was 1.78 × 10^−4^ and 2.93 × 10^−5^, respectively). Moreover, the expression of proto-oncogenic markers *p53*, *p21*, and *c-myc* was determined at (median of 2^−ΔCT^ was 3.02 × 10^−3^, 2.41 × 10^−2^, and 5.39 × 10^−3^, respectively) (Figure 2).

The mRNA expression of the *CD133* marker was determined in the cells derived from post-op. of OvCa (N = 8) and ascitic fluid (N = 9) from 11 patients. For cells from post-op. tissue of OvCa (N = 7), a median value for 2^−ΔCT^ of *CD133* was 2.02 × 10^−4^, while for cells from ascitic fluid (N = 5) was 7.25 × 10^−6^ (Figure 3). For cells from tumor tissue (N = 2) and ascitic fluid (N = 3), the *CD133* expression was not detectable.

#### 2.1.3. Expression of Membrane Markers and Transcription Factors

Immunofluorescence staining was performed on cells derived from 13 patients for the presence of cancer-associated fibroblasts (CAFs) markers (fibroblasts activation protein—FAP, platelet-derived growth factor receptor alpha—PDGFRα), epithelial to mesenchymal transition (EMT) marker (Snail, vimentin), CSCs markers (ALDH1, c-kit), pluripotency-related markers (Oct4, Sox2, Nanog). Moreover, the cytoskeleton organization including F-actin (actin filaments) and vimentin (intermediate filaments) was analyzed.

For high-grade serous OvCa, the cells from post-op. tumor and ascitic fluid were positive for CD44, and weak expression of CAFs (PDGFRα, FAP) was observed. The cells were also positive for pluripotency-related markers (Oct4, Sox2, Nanog) and CSCs markers including ALDH1 and c-kit. The expression of F-actin showed the different organization of the cytoskeleton and the presence of filopodia (reach in F-actin) and the stress fibers along the cells. Moreover, the cells expressed vimentin localized around the cell nuclei. For cystadenofibroma, the cells from post-op. tissue and ascitic fluid were positive for CD44, Oct4, Sox2, Nanog, ALDH1, and c-kit. The cells also expressed F-actin and vimentin. Additionally, as cells from high-grade serous OvCa, cystadenofibroma occasionally present weak expression of CAFs markers (PDGFRα, FAP) (Figure 4).

### 2.2. Internalization of HATMSC2-MVs into Primary Ovarian Cancer Cells

Prior functional tests’ internalization of HATMSC2-MVs into primary OvCa cells has been assessed. Internalization of far-red (DiD)-labeled HATMSC2-MVs into green (PDGFRα)-labeled primary OvCa cells after 24 h of co-culture was analyzed using confocal microscopy imaging. DiD-positive HATMSC2-MVs were present in the nuclei and cytoplasm of primary OvCa cells (Figure 5, right and left image). The image on the left shows a full cell volume (in green) with several DiD-labelled vesicles (in red). The right image demonstrates a single vertical section of the same cell proving cytoplasmic presence of the DiD-positive vesicle.

### 2.3. Effect of HATMSC2-MVs on the Metabolic Activity of Primary Ovarian Cancer Cells

The effect of HATMSC2-MVs on OvCa of metabolic activity, reflecting the proliferation, was determined by MTT assay. The decreased metabolic activity of OvCa cells treated with HATMSC2-MVs at the ratio 100:1 (100 HATMSC2-MVs: 1 OvCa cell) was observed. The untreated cells served as a control. For cells from post-op. tumor of high-grade serous OvCa (N = 9), the median value of relative absorbance at 570 nm on day 0 and day 3 was (0.97 and 0.62, respectively, *p* < 0.01), while for cells from ascitic fluid (N = 8), the median value on day 0 and day 3 was (0.99 and 0.40, respectively, *p* < 0.001). Similar results were observed for cells derived from other histological types of OvCa. For cells from post-op tissue N = 5, the median value on day 0 and day 3 was (0.90 and 0.32, respectively, *p* < 0.001), and for cells from ascitic fluid N = 4, the median on day 0 and day 3 was (0.98 and 0.32, respectively, *p* < 0.001) (Figure 6).

### 2.4. Effect of HATMSC2-MVs on the Migration Activity of Primary Ovarian Cancer Cells

The migration activity of OvCa cells was assessed using a scratch test. The cells were treated with HATMSC2-MVs at a ratio of 5:1 (5 HATMSC2-MVs: 1 OvCa cell), and the dynamic of scratch closure was evaluated at four time-points up to 28 h (Figure 7a,b). HATMSC2-MVs decreased migratory activity of OvCa cells isolated from post-op. tumor of high-grade serous OvCa as confirmed by the median value of relative wound closure assessed for 0.88 (estimated relative to control untreated cells), while for cells from ascitic fluid, the median was at the control level and revealed 1.06. In contrast, HATMSC2-MVs do not exert inhibitory effect on migration of OvCa cells derived from other histological types of OvCa. For cells from post-op tissue, the median value of relative wound closure was 0.99, while for cells isolated from ascitic fluid, the median was 1.16 (Figure 7c). Representative images from microscopic imaging are presented in Figure 7a,b.

### 2.5. Effect of HATMSC2-MVs on the Fate of Primary Ovarian Cancer Cells—Survival and Apoptosis

Immunofluorescence staining of OvCa cells allows assessment of live cells stained with Syto 9 (green), dead cells stained with propidium iodide (red), and HATMSC2-MVs stained with DiD (violet). The images acquired by confocal microscopy were processed, and live to dead channel ratio was calculated with Fiji/ImageJ software version 1.54f. For cells from post-op. tumor of high-grade serous OvCa, treated with HATMSC2-MVs, the median value of the relative live/dead ratio was 0.95, while for cells from ascitic fluid, the median value revealed 1.06 (at the control level). Increased live/dead ratio was observed for cells derived from other histological types of OvCa exposed for HATMSC2-MVs. For cells from post-op tissue, the median value was calculated for 1.29, while for cells from ascitic fluid, the median value revealed 1.22 (Figure 8a). Representative images from confocal microscopy are presented in (Figure 8b,c).

The cell death process was assessed by apoptosis/necrosis assay. The cancer cell survival following treatment with HATMSC2-MVs at the ratio 100:1 was analyzed on cells derived from 12 patients (16 samples) using the flow cytometry method (Figure 9a,b). 

The results showed that the percentage of alive cells after treatment with HATMSC2-MVs increased compared to control (median 67.27% vs. 52.00%) in cancer cells derived from two patients (two post-op. of OvCa). Moreover, the percentage of early apoptotic cells (median 16.38% vs. 21.35%) and late apoptotic cells (median 9.87% vs. 18.61%) following treatment decreased compared to untreated cells. The treatment with HATMSC2-MVs also decreased the percentage of necrotic cells compared to the control (median 6.5% vs. 8.03%).

The percentage of alive cells did not change compared to untreated control (median 67.27% vs. 67.63%) for five samples (two ascitic fluid and three post-op., five patients). Furthermore, the percentage of early apoptotic cells increased in treated group vs. control (median 2.97% vs. 0.95%). In the treated group, the percentage of late apoptotic cells (median 2.88% vs. 5.57%) and necrotic cells (median 16.75% vs. 21.74%) decreased compared to control. 

Importantly, for nine samples (six ascitic fluid and three post-op. of OvCa, seven patients) treated with HATMSC2-MVs, the decrease of percentage of alive cells (median 50.94% vs. 76.25%; *p* < 0.01) was observed. In this group, the median value of early apoptotic cells was assessed for 3.52% vs. 4.56% in the control cells; however, the range of early apoptotic cells in this HATMSC2-MVs-treated group was assessed from 0.2% to 41.45%. The increase of proportion of late apoptotic cells (median 14.98% vs. 8.5%) was observed. Apart from apoptotic pathway, the treatment of cells with HATMSC2-MVs leads to increase of the percentage of necrotic cells compared to untreated cells (median 21.27% vs. 12.20%).

### 2.6. Characteristics of 3D Model of Primary Ovarian Cancer—Spheroids

To assess the effectiveness of the proposed HATMSC2-MVs therapy in a tumor, a 3D model of primary OvCa cells isolated from the tumor and ascitic fluid has been created. OvCa spheroids were characterized by their size and CSCs phenotype. The spheroids were formed from post-op OvCa cells (N = 9), the diameter ranging from 223 μm to 741 μm (a median value 462 μm). For spheroids derived from ascitic fluid cells (N = 9) the diameter was higher compared to spheroids from post-op OvCa cells and was ranging from 345 μm to 1301 μm with a median value of 538 μm (Figure 10a,b). Created spheroids assessed for CSCs markers revealed expression of CD44, CD133, and CD24. Spheroids formed from cells isolated from post-op tissues of OvCa contained a population of CD44 cells ranging from 10.0 to 33% with median value 24%, CD133 (range from 1.0% to 17.0%, median 3.0%), and CD24 (range from 1.0% to 16.0%, median 8.0%). While for cells from ascitic fluid, the range and median were as follows: for CD44 were (6.0–70.0%, median 27.0%), CD133 (1.0–22.0%, median 10.0%), and CD24 (0.5–41.0%, median 5.0%) (Figure 10c).

### 2.7. Effect of HATMSC2-MVs on the Survival of Primary Ovarian Cancer Spheroids

Identification of the impact of HATMSC2-MVs on primary cancer cells in a 3D model of OvCa will allow development of the study on efficient therapeutic strategy. Live/dead assay was applied to assess the inhibitory effect of HATMSC2-MVs on OvCa. Untreated spheroids served as a control. For spheroids from post-op tumor of high-grade serous OvCa, the median value of the relative live/dead ratio calculated to the control spheroids was 0.85, while for spheroids from ascitic fluid, the median value was 1.02 compared to the control. Different results were obtained for spheroids derived from other histological types of OvCa. For spheroids from post-op. tumor, the median value was 1.33, while for spheroids from ascitic fluid, the median was 0.85 (Figure 11a). Representative images from confocal microscopy are presented in (Figure 11b).

### 2.8. Analysis of Bioactive Factors of HATMSC2-MVs Involved in Apoptosis Pathway

The content of HATMSC2-MVs was analyzed for the presence of bioactive factors that regulate apoptosis by using the Membrane-Based Protein Array. The heat-map revealed differences in the expression of analyzed factors between HATMSC2-MVs and parental cells HATMSC2. HATMSC2 cells expressed all analyzed bioactive factors (43 analytes) while HATMSC2-MVs do not express bax protein (Figure 12). In HATMSC2-MVs 15 pro-apoptotic factors (bad, BID, BIM, caspase 3, cytochrome c, Fas, FasL, HTRA, IGFBP-3, IGFBP-5, p27, p53, SMAC, TRAIL-R1, and TRAIL-R2) were detected at a higher level compared to parental HATMSC2 cells (Figure 12b,c). Only CD40, as a pro-apoptotic factor, was detected at a higher level in HATMSC2 cells. On the other hand, the HATMSC2-MVs contained 12 anti-apoptotic cytokines (bcl-2, bcl-w, HSP60, HSP70, IGF-I, IGF-II, IGFBP-6, livin, p21, TRAIL-R3, TRAIL-R4, and XIAP) at a higher level compared to HATMSC2 cells.

## 3. Discussion

Over the years, the important role of extracellular vesicles (EVs) as one of the mechanisms of cell-to-cell communication has been extensively studied [22]. EVs are secreted by all normal cells, cancer cells, and apoptotic cells [23]. The bioactive cargo of EVs consists of cytoplasmic proteins (including enzymes and transcription factors), membrane adhesive molecules, lipids, and nucleic acids (including mRNA and regulatory miRNA). There are different types of EVs interaction with target cells leading to different biological effects, including: (i) direct activation of the target cell surface receptor after binding of the ligand derived from EVs, (ii) transfer of receptors present on the surface of EVs to the recipient cell, and (iii) functional modifications in target cells as a result from the interaction of specific miRNA or proteins transferred in vesicles with cellular effectors present in target cells [22].

There is an increasing interest in the study of MCs and their derivative EVs as new therapeutic options in several research fields including anticancer therapy, due to their role in different biological processes, including cell proliferation, apoptosis, angiogenesis, inflammation, and immune response. The biological potential of EVs is based on the biomolecular cargo transported inside these particles. However, not only composition of bioactive factors of EVs but also EVs from the same source of parental MSC can have opposite effect depending on the type of tumor cells [24,25]. 

A recent study by our research group confirmed that MVs derived from human immortalized MSC of adipose tissue origin (HATMSC2) affect the biological activity of two human ovarian cancer cell lines ES-2 (representing poorly differentiated ovarian clear cell carcinoma) and OAW-42 (representing ovarian cystadenocarcinoma) [17]. We demonstrated that HATMSC2-MVs treatment decreases the proliferation activity of ovarian cancer cells and induces cell death via apoptosis and/or necrosis pathway in MVs dose-dependent manner. These antitumoral effects are possibly associated with changes in the secretion profile of ovarian cancer cells, especially higher production of anti-tumor cytokines (IL-1ra, IL-2Ra, IL-12-p40, IL-12-p70, and IFN-γ) and decreased level of cancer-promoting cytokines (IL-6, IL-8, GRO-alpha, angiopoetin-2, and VEGF) after HATMSC2-MVs treatment [17]. 

These results encouraged us to assess the influence of HATMSC2-MVs on the biological activity of primary OvCa cells isolated from two sources, post-op. tissue and ascitic fluid. To the best of our knowledge, this is the first study where the effect of cell-free therapy, by using HATMSC2-MVs, has been assessed on primary OvCa cells. Before HATMSC2-MVs application, we characterized primary OvCa cells for the immunophenotype of MSCs and CSCs markers. All analyzed cancer cells express similar MSCs markers CD73, CD90, CD105; however, their pattern of expression is different between analyzed primary OvCa cells. Our previous study revealed that application of HATMSC2-MVs on ovarian cancer cell lines OAW-42 and ES-2 had no effect on the expression of CD73, CD90, and CD105 antigens strongly expressed on both cell lines [21]. However, biological features of the examined primary OvCa cells, characterized by high expression of MSCs markers such as CD73 and CD105, involved in tumor progression and metastasis, can serve as a therapeutic target for specific neutralizing antibodies inhibiting ovarian cancer growth and metastasis [26,27]. 

CSCs constitute a specific population of cancer cells and are known as cells resistant to chemotherapy and apoptosis [4,28]. Ovarian CSCs, characterized here by CD24, CD44, CD133 expression, were detected in the majority of analyzed OvCa samples; all of them express CD44 and CD133, but CD24 was not detected in 4 out of 25 analyzed samples (from tumor and ascitic fluid). CD133 is widely used for CSCs isolation in different solid tumors. In the case of ovarian cancer tumors, it has been reported that elevated levels of CD44- and CD133-positive cells were associated with chemotherapy-resistant patients and were present in metastatic and recurrent tumors (reviewed in [4]). This is in line with different reports that CSCs markers are not specific for all ovarian cancers, but their expression can predict tumor aggressiveness [29,30]. In this study, we confirmed the presence of OvCa cells with the same phenotype as tumor cells in the corresponding ascitic fluid that can be helpful in diagnosis and therapeutic approach design before surgical procedures.

Evaluation of pluripotency-related markers (Oct4, Sox2, Nanog) and proto-oncogenic transcripts (*p53*, *p21*, *c-myc*) has been performed to assess biological feature of examined OvCa cells. Oct4, Sox2, and Nanog are known as transcription factors involved in maintaining pluripotency and self-renewal [31]. Oct4 plays a role in the regulation of several signaling pathways in CSCs such as Hedgehog, Wnt, Notch, PI3K/Akt, and JAK/STAT [32]. In our study, we confirmed the expression of Oct4, Sox2, and Nanog in primary OvCa cells from both sources, ascitic fluid and post-op. tissue, by RT-PCR and/or immunofluorescence staining. Enhanced expression of Oct4, Sox2, and/or Nanog are characteristic for primary OvCa tumors, and their expression enhances chemoresistance and potential of tumor relapse [31]. Oct4, Sox2, and Nanog can be a target for cancer treatment. The knockdown of Oct4 led to the inhibition of proliferation activity, migration activity, and invasiveness of two ovarian cancer cell lines A2780 and SKOV-3, and inhibited the tumor growth and metastasis in a mouse model [33]. High expression of Sox2 is related to poor prognosis of cancer treatment. It was demonstrated that higher expression of Sox2 was detected by immunostaining in metastasis specimens compared to primary tumor [34]. The silencing of Sox2 caused decreased spheroids formation by SKOV3 and HO8910 cells and expression of EMT and stem cell-related genes, and similarly to Oct4 leads to inhibition of proliferation and migration of cells [35]. In ovarian cancer, Nanog affects cell migration and invasiveness. For the survival of patients with ovarian serous carcinoma, positive Nanog expression was lower compared to patients with negative Nanog. Thus, Nanog can be used as a predictive marker of ovarian cancer patients’ survival [36]. Our studies showed overexpression of the proto-oncogene *c-myc* in the examined OvCa cells. *c-myc* regulates the expression of genes involved in cell proliferation, differentiation, apoptosis, and stem cell renewal. These properties are attractive as a therapeutic target in many tumors including OvCa. In high-grade serous OvCa, the overexpression of *c-myc* is associated with resistance to platinum-based therapy and decreased overall survival. Targeting *c-myc* by siRNA-mediated *c-myc* silencing of cisplatin-resistant A2780 OvCa cell line decreased cell proliferation and survival and inhibited cell-cycle progression [37].

In our study, the immunophenotype of primary OvCa cells has been expanded to include the expression of factors related to markers for CAFs (FAP, PDGFRα), epithelial to mesenchymal transition (EMT) markers, Snail and vimentin, and CSCs markers (ALDH1, c-kit). FAP and PDGFRα are known as cancer-associated fibroblasts markers [38]. In our study, the FAP and PDGFRα revealed weak expression or were undetected and this suggests that CAFs were poorly represented in the population of primary OvCa cells. One of the key biomarkers of EMT is vimentin, a type III intermediate filament that is normally expressed in mesenchymal cells but is upregulated during cancer metastasis. [39]. Vimentin intermediate filaments modulate Snail, which is a main transcription factor involved in EMT [39]. The inhibition of Snail via miR-137 and miR-34a leads to the inhibition of the production of Snail on transcription and protein levels and decreases the invasiveness and spheroids formation in ES-2 and SKOV-3 cell lines [40]. The presence of ALDH-1-positive cells in ovarian cancer patients leads to poor prognosis of survival [41]. The silencing of ALDH-1 leads to inhibition of tumor growth in mouse models. Increased expression of c-kit (CD117) in ovarian cancer is involved in chemoresistance. Phenotypic diversity in ovarian cancer tumors is challenging for therapeutic approaches, especially when different phenotypes of CSCs are presented and targeting therapy for one molecule will be ineffective.

In this study, we propose cell-free therapy with HATMSC2-MVs, which can be applied as a supportive biotherapy procedure in OvCa treatment. Our results revealed that the application of HATMSC2-MVs treatment on primary OvCa cells leads to a decrease in the metabolic activity of OvCa cells and apoptosis induction. The difference in metabolic activity of treated high-grade serous OvCa cells from both sources such as tumor and ascitic fluid, compared to the control untreated cells, is the most visible on day 3, while for the cells from the other histological types of ovarian cancer on day 2. Decrease of OvCa cells proliferation after exposure to HATMSC2-MVs leads to decreased migratory activity of OvCa cells isolated from post-op. tumor as confirmed in a scratch test. This phenomenon suggests that HATMSC2-MVs content may inhibit aggressiveness and invasion of OvCa cells and thus play a role as regulator of tumor progression. 

In a previous study, we demonstrated that HATMSC2-MVs treatments are involved in the regulation of cell cycle and apoptosis induction of the ovarian cancer cell lines OAW-42 and ES-2 [21]. In the present study, we confirmed that HATMSC2-MVs promote apoptosis and contribute to necrosis in the majority of tested primary OvCa cells, and this may lead to tumor attenuation. To date, the knowledge about the exact mechanisms of MSC secretome-mediated effects on cancer cells is limited; however, data so far suggest complex regulation involving antagonistic effects [42]. 

To determine a bioactive cargo of HATMSC2-MVs that can induce cell death, we analyzed the proteins content involved in apoptosis signaling by using the RayBio^®^ Human Apoptosis Antibody Array. Our results showed that the HATMSC2-MVs contain several pro-apoptotic proteins (including bad, BIM, caspase 3, Fas, FasL, IGFBP-5, p27, p53, TRAIL-R1, and TRAIL-R2) and anti-apoptotic factors (including bcl-2, HSP-60, IGF-I, IGFBP-6, p21, TRAIL-R3, and TRAIL-R4) at a higher level compared to HATMSC2 cells. In this study, HATMSC2-MVs serve as an apoptotic signal delivery system into cancer cells as confirmed by internalization of HATMSC2-MVs into primary OvCa cells by confocal microscopy imaging. HATMSC2-MVs-internalized OvCa cells release pro-apoptotic and anti-apoptotic molecules, and their activity depends on many factors influencing apoptotic signal induction or repression that may have a variety of outcomes depending on the molecular or transcriptional context.

Tumor necrosis factor (TNF)-related apoptosis-inducing ligand (TRAIL) belongs to the TNF family and is capable of inducing apoptosis in cancer cells while sparing normal cells. In cancer cells, TRAIL exerts its apoptotic effects through activation of caspase 8, which triggers apoptosis cascade by binding to the death receptors TRAIL-R1 and TRAIL-R2 on target cancer cells [43]. Research performed by Shamili et al. [44] showed that the exosomes derived from genetically modified MSCs to express TRAIL can delay the tumor growth and cause the decrease of tumor volume in a melanoma mouse model [44]. Moreover, the expression of TRAIL in MSCs can be induced via X-ray radiation [45]. Studies on MSC-derived EVs, as potential drug delivery vehicles, showed that TRAIL-transduced MSCs secrete EVs expressing surface TRAIL molecules that are efficiently induced apoptosis in different 11 cancer cell lines but they are not cytotoxic for normal cells [46]. Our study confirmed that MVs from immortalized HATMSC2 cell lines express the death receptors TRAIL-R1 and TRAIL-R2, which contribute to apoptosis induction. TRAIL also functions in the form of a soluble ligand and can trigger p53-independent apoptosis. A recent study demonstrated that genetically engineered umbilical cord MSCs by using recombinant TRAIL (rhTRAIL) have been designed to continuously express and secrete soluble TRAIL (MSC-sTRAIL) as a potential therapeutic approach for treatment of hematologic disorder B-cell acute lymphocytic leukemia (B-ALL). Studies performed in vitro and in vivo on the mouse B-ALL model revealed that MSC-sTRAIL can inhibit B-ALL cell proliferation by initiating apoptosis via the caspase cascade-mediated pathway and mitochondrial-mediated pathway [47].

Our study revealed upregulated expression of proapoptotic caspase 3 in HATMSC2-MVs compared to parental cells. Earlier studies on the antitumor effect of MSCs of bone marrow origin on mouse hepatoma cells documented that MSCs could induce inhibitory effects on tumor cells via upregulation of mRNA expression of cell cycle negative regulator p21 and pro-apoptotic protease caspase 3, leading to apoptotic death of cancer cells [48]. In the present study, we demonstrated that HATMSC2-MVs contain high levels of proapoptotic caspase 3 and high levels of cell cycle negative regulator p21. The p21 represent cyclin-dependent kinase inhibitor and is a major player in cell cycle control [49]. The suppressor function of p21 against cancer cells is associated with the ability of p21 to inhibit cell proliferation through the inhibition of CDK2 activity and cell growth arrest at G0/G1 phase. However, a number of oncogenes inhibit the activity of p21 to promote cell growth and tumorigenesis. As reported, p21 also functions as an inhibitor of apoptosis, thus p21 repression may have an antitumor effect by sensitization of cancer cells to apoptosis by anticancer drugs [49]. The differential effects of p21 on cell cycle arrest and apoptosis are also account by the activity of caspase 3, which specifically cleavage p21 during the DNA damage-induced apoptosis of cancer cells [50]. These studies showed that caspase-3-mediated cleavage and repression of the p21 protein may change cancer cells from growth arrest to undergoing apoptosis. Based on these observations we suggest that HATMSC2-MVs cargo, where both pro-apoptotic caspase 3 and anti-apoptotic p21 were upregulated, can influence OvCa cells’ apoptosis or growth arrest after internalization into OvCa because the cleaved p21 could not arrest the cells in the G1 phase nor suppress the cells undergoing apoptosis; however, this hypothesis should be verified in further studies. 

Finally, assessment of HATMSC2-MVs efficacy was performed in a 3D model of spheroids composed from primary OvCa cells. Spheroids are widely used as a 3D model to determine the cytotoxicity of newly developed drugs especially in cancer treatment. In our study, we used spheroids to determine the effect of HATMSC2-MVs on the survival of the cells in a 3D model mimicking ovarian tissue. Our model is composed from a heterogeneous population of OvCa cells including CSCs with CD133, CD44, and CD24 phenotype. The presence of CD44 is associated with metastatic and recurrence capacity of ovarian cancer and poor overall survival [4]. CSCs in OvCa expressing CD133 and CD24 are chemoresistant, and spheroids with the presence of these markers can serve as a model for therapeutic approaches in regulation of CSCs markers expression because targeting one marker alone may be not sufficient to achieve the expected anti-cancer effect [4]. Internalization of HATMSC2-MVs has been confirmed, and live/dead assay revealed a decreased ratio of alive cells in OvCa isolated from ovarian tumor tissue. The study performed by Liao et al. showed that the spheroids derived from primary ovarian cancer cells are more sensitive to cisplatin compared to cells in the 2D model due to high proliferation activity [51]. On the other hand, the spheroids derived from the MCW-OV-SL-3 (derived from endometrioid ovarian cancer) cell line are resistant to cisplatin due to increased activation of ERK and PI3K/AKT signaling pathway [52]. A recent study demonstrated that patient-derived organoids formed from cells isolated from ascitic fluid were successfully used as a reliable model for assessment of molecular mechanisms underlying platinum resistance to extend the understanding in treatment of gynecological serous carcinomas [53].

Overall, HATMSC2-EVs could exert either antiproliferative or proapoptotic effects on OvCa cells; however, different strategies to employ MSCs and their derivatives as a biotherapy in ovarian cancer are studied. Melzer et al. [54] showed that taxol-loaded microvesicles did not affect the viability of analyzed cancer cell lines including ovarian adenocarcinoma (SKOV-3). However, taxol-loaded exosomes reduced viability and proliferation activity of human ovarian adenocarcinoma similarly compared to taxol, but the concentration of taxol in the exosomes was lower [54].

Moreover, EVs released by MSCs of bone marrow or adipose tissue origin exhibit suppressive effects on ovarian cancer cell lines. Anti-proliferative and anti-tumor activity of MVs produced by BM-MSCs on the SKOV3 cell line as well as in an SCID mice model was reported [16]. In the animals, the presence of necrotic areas in the tumor section was confirmed by histological analysis [16]. Similar results were observed by Wang et al. [55]. Extracellular vesicles derived from MSC decreased proliferation, migration, and invasiveness of ES-2 and ovarian adenocarcinoma (CAOV-3) cell lines. Moreover, this group discovered that in patients with OvCa, miR-18a-5p is down-regulated and associated with poor prognosis of survival. Thus, the application of EVs with overexpression of miR-18a-5p caused an increase in cisplatin sensitivity of analyzed ovarian cancer cell lines. 

One of the mechanisms of EVs’ action is non-coding RNAs, principally miRNAs. The study performed by Reza et al. [56] reported that miRNAs delivered in AT-MSCs derived exosomes are responsible for inducing particular cell signaling pathways in target ovarian cancer cells. The co-incubation of proteinases digested exosomes with epithelial ovarian cancer cells A2780 and SKOV3 cells leads to a reduction of cancer cell viability, colony formation ability, and induction of apoptosis. Most of the miRNAs detected in AT-MSCs-derived exosomes were mainly responsible for catalytic activity, channel regulatory activity, and metabolic and cellular processes.

Despite numerous analyses, we note that the presented studies have some limitations. In these studies, we performed a broad characterization of the phenotypic and molecular profiles of primary OvCa cells; however, an attempt to correlate them with the response toHATMSC2-MVs treatment did not yield significant results. This is probably due to the heterogeneous population of primary ovarian cancer cells that expressed analyzed markers at different levels and with diverse numbers of samples in each group. For this reason, it would be beneficial to perform further experiments on OvCa cells derived from greater numbers of patients. Another limitation is the need to better determine the biological cargo of HATMSC2-MVs. Further research, especially the whole proteomic analysis of HATMSC2-MVs by liquid chromatography mass spectrometry (LC-MS) and transcriptomic analysis by next-generating sequencing (NGS), could be useful to allow for identification and quantification of both unknown and known bioactive compounds of HATMSC2-MVs. Moreover, it would be important to examine which bioactive factors caring by HATMSC2-MVs are critical for the determination of ovarian cancer sensitivity. Neutralizing antibodies or siRNA for some factors of interest would be preferred for these studies. However, despite these limitations, the present study adds a novel observation for HATMSC2-MVs biological characteristics and their potential ability to inhibit primary OvCa cells growth and apoptosis induction; however, further studies are needed to clarify their anti-cancer activities.

## 4. Materials and Methods

### 4.1. Isolation and Characterization of Human Primary Ovarian Cancer Cells

The primary ovarian tumor cells were isolated from two sources: human postoperative tissue of OvCa and ascitic fluid (16 patients), according to the permission from the Bioethics Committee at the Medical University of Wroclaw (No. KB-489/2020). Procedures were performed at the Department of Gynecology and Obstetrics, Wroclaw Medical University. Characteristics of patients are presented in Table 1. 

In our study, 62.7% of OvCa was high-grade serous ovarian cancer (10 patients). Six patients were classified to other histological types of OvCa (37.5%). That group consists of ovarian clear cell carcinoma (N = 2), cystadenofibroma (N = 1), mucinous ovarian cancer (N = 2), and malignant mixed Mullerian tumor (MMMT), also called a carcinosarcoma (N = 1). Tumor cells were isolated using Tumor Dissociation Kit Human (Milteni Biotec, Bergisch Gladbach, Germany) according to the manufacturer’s instructions. Briefly, the tissues were cut into small pieces and then transferred into a gentleMACS Tube dedicated to gentleMACS Dissociator (Milteni Biotec, Bergisch Gladbach, Germany) containing the mixture of Enzyme A, H, and R. Next, the tissues were proceeded in triple steps by mechanical and enzymatic digestion using the gentleMACS Dissociator. Between the digestion steps, the samples were incubated for 30 min at 37 °C with mixing three times. Finally, samples were transferred into a 70 µm cell strainer. Cell suspensions were centrifuged at 300× *g* for 7 min, and then the cells were resuspended in medium. The cells were cultured in the DMEM (IIET, Wroclaw, Poland) and OptiMEM GlutaMax media (Thermo Fisher Scientific, Carlsbad, CA, USA), mixed in equal proportions. The DMEM medium was supplemented with 10% FBS (Thermo Fisher Scientific, Carlsbad, CA, USA), a 1% penicillin/streptomycin solution (Thermo Fisher Scientific, Carlsbad, CA, USA), and 1% L-glutamine (Thermo Fisher Scientific, Carlsbad, CA, USA). The OptiMEM GlutaMax medium was supplemented with 3% FBS and a 1% penicillin/streptomycin solution. Tumor cells from the ascitic fluid were isolated according to the protocol described by [57]. Briefly, the ascitic fluid was centrifuged at 1000× *g* for 10 min. The obtained cells were washed twice in PBS (IIET, Wroclaw, Poland) and finally resuspended in culture medium. The cells were cultured in the DMEM supplemented with 10% FBS, a 1% penicillin/streptomycin solution, and 1% L-glutamine.

### 4.2. The Characterization of Human Primary Ovarian Cancer Cells

#### 4.2.1. Analysis of Membrane Markers by Flow Cytometry

The characterization of primary OvCa cells included flow cytometry analysis of the expression of MSCs markers (CD73, CD90, CD105), CSCs markers (CD44, CD24, CD133), hematopoietic stem and progenitor marker (CD34), and leukocyte common antigen (CD45).

For flow cytometry analysis, the primary ovarian tumor cells were detached from cell culture flasks using the Tryple Select (Thermo Fisher Scientific, Carlsbad, CA, USA) solution. Afterward, the cells were incubated at 4 °C for 30 min with PE-conjugated antibodies for the human CD73, CD90, CD105, CD44, CD45, CD133 (BD Biosciences, San Jose, CA, USA), and FITC-conjugated CD24, CD34 antibody (BD Biosciences, San Jose, CA, USA), and with the corresponding isotype controls (BD Biosciences, San Jose, CA, USA). Then, the PBS (IIET, Wroclaw, Poland) was utilized for cell washing. The BD LSRFortessa flow cytometer (BD Biosciences, San Jose, CA, USA) was used to analyze the labeled cells. The data were acquired using the CellQuest software version 5.1 (BD Biosciences, San Jose, CA, USA). The histograms were prepared using Flowing Software 2 (Perttu Terho, Turku Centre for Biotechnology, Turku, Finland).

#### 4.2.2. Analysis of Transcripts by Real Time RT-PCR

The genetic analysis of cells by real-time RT-PCR were used to determine the level of transcripts involved in the maintenance of pluripotency (Oct4, Sox2), pro-oncogenic transcripts (p53, p21, and c-myc), and CD133. RNA isolation was performed using a NucleoSpin^®^ RNA Kit (Macherey-Nagel, Düren, Germany), then the reverse transcription PCR was performed according to the manufacturer’s instruction of RevertAid First Strand cDNA Synthesis Kit (Thermo Fisher Scientific, Weston, FL, USA). The level of transcripts was determined using the real-time PCR method with the TaqMan Master Mix (Thermo Fisher Scientific, Weston, FL, USA) and specific TaqMan probes: Sox2 (Hs01053049), Oct4 (Hs00999632), p53 (Hs00153349), c-myc (Hs00153408), CD133 (Hs01009250), and GAPDH (Hs03929097) (Thermo Fisher Scientific, Weston, FL, USA). Quantification of mRNA content was performed on the ViiA 7 Real-Time PCR System (Applied Biosystems, Foster City, CA, USA). The calculation of results was performed using the ΔCt method [58], and the GADPH gene was applied as reference. 

#### 4.2.3. Analysis of Membrane Markers and Transcription Factors by Fluorescence Microscopy Imaging

Immunofluorescence staining of the cells for microscopic imaging included the following markers: CD44, F-actin, vimentin, cancer stem cells markers (ALDH1, c-kit (CD117)) cancer-associated fibroblasts markers (PDGFRα, FAP), epithelial to mesenchymal-transition markers (Snail, vimentin), and pluripotency-related markers (Oct4, Sox2, Nanog). The cells were fixed with 4% PFA (Merc, Kenilworth, NJ, USA) (15 min in room temperature (RT)) and permeabilized with 0.1% Triton X100 (Avantor Performance Materials Poland, Gliwice, Poland) (15 min in RT). Then, the unspecific binding of antibodies was blocked using 1% BSA (Symbios, Gdansk, Poland) (40 min incubation, RT). Selected primary antibodies were applied for 1 h (RT) or overnight incubation (4 °C) (Appendix A). Then, the cells were labeled for 45 min (RT) with Alexa Fluor 488-conjugated goat anti-mouse IgG and/or Alexa Fluor 647-conjugated goat anti-rabbit IgG (Invitrogen, Carlsbad, CA, USA). The actin filaments were stained for 45 min (RT) with Alexa Fluor 488-conjugated phalloidin (Invitrogen, Carlsbad, CA, USA) (Appendix A), and the cell nuclei were stained for 30 min (RT) with NucBlue™ Fixed Cell ReadyProbes™ Reagent (DAPI) (Invitrogen, Carlsbad, CA, USA). Image acquisition was performed with an Axio Imager Z2 microscope (Zeiss, Gottingen, Germany) equipped with 40× (NA 0.75), dry objective using Zen 2.6 Blue edition Software (Zeiss, Gottingen, Germany). The images were processed with the Zen 2.6 Blue edition Software.

### 4.3. Isolation and Characterization of MVs Derived from HATMSC2 Cells

HATMSC2-MVs were isolated from conditioned media harvested from HATMSC2 cell cultures using the sequential centrifugation-based method according our well-establish method [21]. Briefly, the conditioned media were collected from HATMSC2 cells cultured for 48 h under hypoxic condition at 1% O_2_ in serum-free medium DMEM supplemented with P/S and L/G. The conditioned media were centrifuged twice at 4 °C for 10 min at a speed of 300× *g* and 2000× *g*, respectively. Then, the conditioned media were centrifuged at 4 °C for 30 min at a speed of 12,000× *g* using a Sorvall LYNX 6000 ultracentrifuge (Thermo Scientific, Carlsbad, CA, USA). After, the PBS were used to resuspend MVs. The MVs were characterized according to Minimal Information for Studies of Extracellular Vesicles (MISEV) recommendation [59]. The size distribution of MVs was analyzed by DLS (Malvern Zetasizer, Malvern, UK), and the average size of MVs was assessed for 460 nm ± 60 nm (Appendix A). Moreover, the MVs were analyzed by transmission electron microscope (JEOL, Peabody, MA, USA) (Appendix A). The MVs concentration was determined using a BD Fortessa Flow Cytometer (BD Biosciences, San Jose, CA, USA) and fluorescent counting beads (Count Bright™ Absolute Counting Beads for flow cytometry, Thermo Scientific, Carlsbad, CA, USA). During sample acquisition, 5000 counting beads were collected by BD Fortessa Flow Cytometer (BD Biosciences, San Jose, CA, USA) (Appendix A). The concentration of MVs was determined using the formula included in the instructions for the CountBright^TM^ Absolute Counting Beads, and it was assessed for 1028 MVs/µL. 

### 4.4. Internalization of HATMSC2-MVs into Human Primary Ovarian Cancer Cells

Internalization of DiD-labelled HATMSC2-MVs into human primary OvCa cells was analyzed using a confocal microscope. Primary cells were stained in a suspension for the expression of PDGFRα. The cells were fixed with 4% PFA (15 min in RT) and permeabilized with 0.1% Triton X100 (0.1%; 15 min in RT). Then, the unspecific binding of antibodies was blocked using 1% BSA (40 min incubation, RT). Mouse anti-human PDGFRα antibody was applied for 1 h (RT). Then, the cells were labeled for 45 min (RT) with Alexa Fluor 488-conjugated goat anti-mouse IgG (Invitrogen, Carlsbad, CA, USA). The cell nuclei were stained for 30 min (RT) with NucBlue™ Fixed Cell ReadyProbes™ Reagent (DAPI) (Invitrogen, Carlsbad, CA, USA). Image acquisition was performed with a Leica SP8 MP confocal microscope (Leica Microsystems, Wetzlar, Germany) equipped with a 40× (NA 1.3) oil objective and spectral detectors using 400 nm, 488 nm, and 638 nm lasers sequentially. The images were processed with Imaris software version x64 9.5.1 (Bitplane, Zurich, Switzerland) to visualize HATMSC2-MV presence within OvCa cells.

### 4.5. Assessment of HATMSC2-MVs Action on the Survival of Human Primary Ovarian Cancer Cells in 2D Culture

#### 4.5.1. MTT Assay

The metabolic activity of cells treated with MVs at a ratio of 100:1 (100 MVs per 1 cell) was analyzed by MTT assay at a different time point (day 0, day 1, day 2, day 3). The cells were seeded in a 96-well plate at the density of 3 × 10^3^ cells/well. The Wallac 1420 Victor2 Microplate Reader (Perkin Elmer, Waltham, MA, USA) was used to determine the absorbance of dye accumulated in cells. The absorbance was measured at 570 nm. 

#### 4.5.2. Scratch Test

Migration activity of primary OvCa cells was assessed using a scratch test. The cells were seeded in a 48-well plate at the density of 5 × 10^4^ cells/well. When the cells were fully confluent, the scars were made with thin tips. The cells were treated with HATMSC2-MVs at a ratio of 5:1 (5 MVs per 1 cell). This ratio was selected based on our previously performed experiments where the ratios of MVs were determined. Since 100 MVs as well as 10 MVs added per 1 ovarian cell were making the visualization on the microscope impossible, the ratio of 5 MVs to 1 ovarian cell was therefore chosen.

Migration was observed at 37 °C in an incubation chamber (PeCon GmbH, Erbach, Germany) with 21% O_2_, 5% CO_2_ mounted on an Axio Observer inverted microscope equipped with a dry 5× (NA 0.16) objective (Zeiss, Gottingen, Germany). The movement of the cells was time-lapse recorded for 2 days at intervals of 4 h using Zen 2.6 Blue Edition Software (Zeiss, Gottingen, Germany). Wound closure was analyzed using Zen Blue Software. Relative wound closure (RWC) was calculated as previously described [60].

#### 4.5.3. Live/Dead Assay in 2D Culture

For microscopic imaging, cells were seeded in a 96-well plate at a density of 3 × 10^3^ cells/well. MVs were labeled with DiD dye (Thermo Fisher, Carlsbad, CA, USA) and incubated with 2D cultures for 72 h. Next, cells were stained for 30 min in RT with propidium iodide (PI) and SYTO 9 according to the manufacturer’s recommendations (LIVE/DEAD kit, Thermo Fisher, Carlsbad, CA, USA), then washed with PBS and fixed with 4% PFA for 15 min. The microscopic imaging was performed using a Zeiss Cell Observer SD confocal microscope equipped with a dry 10× (NA 0.3) objective (Zeiss, Gottingen, Germany) and Rolera EM-C2 camera (QImaging, Surrey, BC, Canada). The labeled cells were detected sequentially using a 488 nm laser (SYTO 9-labelled living cells) and a 561 nm laser (PI-labelled dead or apoptotic cells), while MVs were detected with a 639 nm laser, using 520/35, 600/50, and 690/50 emission filters, respectively. Mosaics composed of 3 × 3 fields of view were recorded as volumes with a 5 μm interval in Z axis. The images were processed with Fiji/ImageJ software version 1.54f (National Institutes of Health, Bethesda, MD, USA). Background noise was removed with median filtering followed by maximum intensity projection (MIP) of acquired channels. To measure the effect of MVs on cells, a SYTO 9 to PI ratio channel was created using an image calculator function by dividing SYTO 9 pixel intensities by corresponding PI pixel intensities at the same coordinates. Next, SYTO 9 and PI MIP images were added using the image calculator function, and the obtained new image was thresholded to reveal cell bodies. The mean SYTO 9 to PI ratio in detected cells was calculated with an analyze particles function.

#### 4.5.4. Apoptosis Assay

After incubation with MVs for 72 h (at the ratio 100 MVs to 1 ovarian cell), the Annexin V and propidium iodide was used for cells staining according to the manufacturer’s instructions (eBioscience™ Annexin V Apoptosis Detection Kits, Thermo Fisher, Carlsbad, CA, USA). The analysis of four populations of cells—alive (Annexin V-negative and propidium iodide-negative), early apoptotic (Annexin V-positive and propidium iodide-negative), late apoptotic (Annexin V-positive and propidium iodide-positive), and necrotic cells (Annexin V-negative and propidium iodide-positive)—was performed using a BD LSRFortessa flow cytometer (BD Biosciences, San Jose, CA, USA). The data analysis was performed by Flowing Software 2 (Perttu Terho, Turku Centre for Biotechnology, Turku, Finland).

### 4.6. Assessment of HATMSC2-MVs Action on the Survival of Human Primary Ovarian Cancer Cells in 3D Culture

#### 4.6.1. Spheroids Formation and Flow Cytometry Analysis

For spheroids formation, cells were seeded in a dedicated 96-well plate Nunclon Sphera (Thermo Fisher, Carlsbad, CA, USA), at the density of 3 × 10^3^ cells/well. To obtain cell suspension for flow cytometry, spheroids were enzymatically digested with Accutase Cell Detachment Solution (Corning, Manassas, VA, USA) and shaken for 10 min at 37 °C. After one washing in PBS, the cells were stained with PE-conjugated antibodies specific for the human CD44, CD133 (BD Biosciences, San Jose, CA, USA), and FITC-conjugated CD24 (BD Biosciences, San Jose, CA, USA) and with the appropriate isotype controls (BD Biosciences, San Jose, CA, USA) for 30 min at 4 °C. Afterward, the labeled cells were washed with PBS (IIET, Wroclaw, Poland) and proceeded using a BD LSRFortessa flow cytometer (BD Biosciences, San Jose, CA, USA). The analysis was performed using Flowing Software 2 (Perttu Terho, Turku Centre for Biotechnology, Turku, Finland).

#### 4.6.2. Live/Dead Assay in 3D Culture

The effect of HATMSC2-MVs at a ratio 100:1 after 72 h of treatment on cancer cell survival in spheroids was analyzed by confocal microscopy imaging. For imaging, HATMSC2-MVs were additionally stained with DiD dye to assess the internalization of MVs into the cancer cells that formed spheroids. After incubation with HATMSC2-MVs for 72 h, the spheroids were stained with propidium iodide and Syto 9 according to the manufacturer’s recommendations (Thermo Fisher, Carlsbad, CA, USA) for 1 h in RT then washed with PBS and fixed with 10% formalin for 30 min. Spheroids were visualized using a Zeiss Cell Observer SD confocal microscope equipped with a dry 10× (NA 0.3) objective (Zeiss, Gottingen, Germany) and Rolera EM-C2 camera (QImaging, Surrey, BC, Canada). The microscopic imaging and image analysis were performed as described in Section 4.5.3.

### 4.7. Examination of HATMSC2-MVs Content to Dissect Apoptotic Factors—Protein Membrane Analysis

The protein content of HATMSC2 cells and HATMSC2-MVs was examined using the Membrane-Based Antibody Array (RayBio^®^ Human Apoptosis Antibody Array C1, RayBiotech, Peachtree Corners, GA, USA). Isolated MVs and cells were lysed in RIPA buffer with a protein inhibitor cocktail for 10 min on ice, sonicated for 15 min, then suspended in PBS. The BCA assay (Thermo Scientific, Carlsbad, CA, USA) was used according to the manufacturer’s instructions to determine the protein concentration of HATMSC2 cells and HATMSC2-MVs. The HATMSC2 cells and HATMSC2-MVs samples were incubated with BCA working reagent for 30 min at 37 °C. The absorbance at 560 nm was measured using the Wallac 1420 Victor2 Microplate Reader (Perkin Elmer, Waltham, MA, USA). The protein concentration of HATMSC2 cells was assessed at 2238 μg/mL while for HATMSC2-MVs was 225 μg/mL. The 50 μg of lysed HATMSC2 cells and HATMSC2-MVs were incubated on a protein membrane according to the manufacturer’s instructions. Briefly, 2.0 mL of blocking buffer was applied on the membrane and incubated 30 min at room temperature. Then, 1.0 mL of HATMSC2 cells and cell-derived microvesicles were incubated with a membrane overnight at 4 °C. Following a series of washes, a biotinylated antibody cocktail was applied on the membrane and incubated for 2 h at RT. The unbound antibody was removed by a series of washes, and the membrane was placed in HRP–streptavidin and incubated for 2 h at RT. Following a third series of washes, chemiluminescent detection was performed, and bound proteins were visualized using X-ray film. A comparison of signal intensities was performed using ImageJ software version 2.9.0 (MosaicJ, Philippe Thevenaz) where relative differences in the expression levels of each analyzed sample were measured and normalized to the intensities of positive control using the Protein Array Analyzer plugin. Automatic analysis of obtained data was calculated using Microsoft^®^ (Office 16) Excel-based Analysis Software Tool for Human Apoptosis kit. The results were calculated as a percentage of expression, where positive control was set to 100% and negative control was set to 0% (relative expression). The cutoff line was set to 5%. All results above 5% were considered real expressions.

### 4.8. Statistical Analysis

All graphs were prepared using GraphPad Prism version 7 (GraphPad Software Inc., San Diego, CA, USA). The statistical analysis was performed using the Mann–Whitney U test and one-way analysis of variance (ANOVA) followed by multiple comparison procedures (Dunnet’s test). A significantly different value was considered when *p* < 0.05.

## 5. Conclusions

In this study, we demonstrate the possibility to inhibit OvCa cells growth and apoptosis induction after exposure of OvCa cells on HATMSC2-MVs treatment. MVs isolated from adipose tissue induced the ovarian cancer cell death in 2D and 3D cellular models. The Membrane-Based Protein Array revealed differences in the expression of analyzed factors between HATMSC2-MVs and parental cells HATMSC2. The present study adds novel observations for potential activity of HATMSC-MVs on primary OvCa cells. HATMSC-MVs with high probability activate the apoptosis process in the target ovarian cancer cells because they are able to release a number of pro-apoptotic factors (e.g., bad, BIM, caspase 3, p27, p53, and others), and this process does not allow for spontaneous repair of damaged cancer cells despite the simultaneous release of anti-apoptotic proteins (e.g., bcl-2, HSP-60, p21, and others) by HATMSC2-MVs. In this study, we used unmodified HATMSC2-MVs that carries bioactive factors known as factors involved in apoptosis. The results suggest that the anti-cancer effect of HATMSC2-MVs is mainly contributed by delivery of molecules that induce cell cycle arrest and apoptosis (p21, tumor suppressor p53, executor caspase 3) and proapoptotic regulators (bad, BIM, Fas, FasL, p27, TRAIL-R1, TRAIL-R2). However, to determine which bioactive factors released by HATMSC2-MVs are critical for the ovarian cancer cell sensitivity to HATMSC2-MVs, further studies by using the neutralizing antibodies or siRNA need to be performed. Moreover, modification of parental MSCs and subsequent application of their derivates in a form of MVs, EVs, or conditioned medium can enhance anti-cancer effects and may serve as a biotherapy, supporting standard procedures.

## Figures and Tables

**Figure 1 ijms-24-15862-f001:**
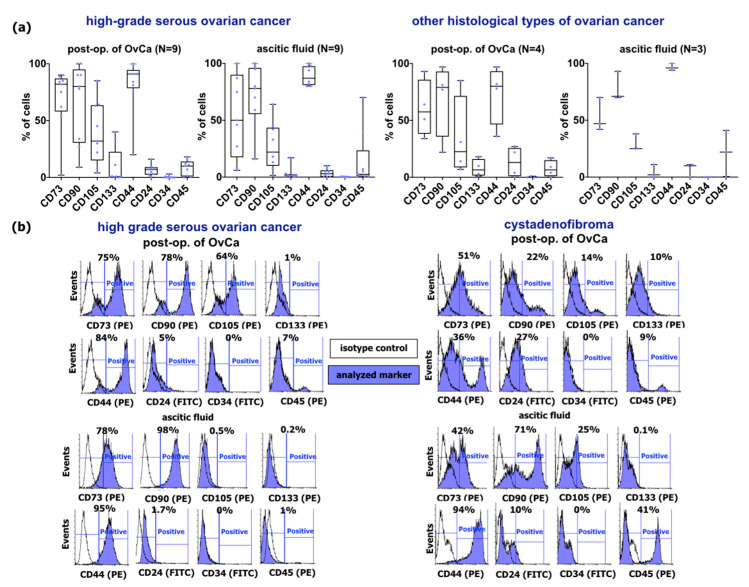
Characteristics of primary OvCa cells from post-op. of OvCa and ascitic fluid. (**a**) Box and whiskers plots show the expression of selected markers by cells from all patients. Cells were stained with selected antibodies conjugated with fluorochromes. Blue dots represent cells from each analyzed patient. Whiskers show minimal and maximal values for each marker. Lines represent the median. The left panel shows plots for cells from high-grade serous OvCa, and the right panel shows plots for cells from other histological types of OvCa. (**b**) Representative histograms of flow cytometry analysis of high-grade serous OvCa (**left** panel) and cystadenofibroma (**right** panel) of OvCa cells from post-operative tumor and ascitic fluid. Blue-filled histograms correspond to cells labeled with defined antibodies, and empty histograms represent the isotype controls. Abbreviations: post-op. of OvCa-post-operative tissue of ovarian cancer.

**Figure 2 ijms-24-15862-f002:**
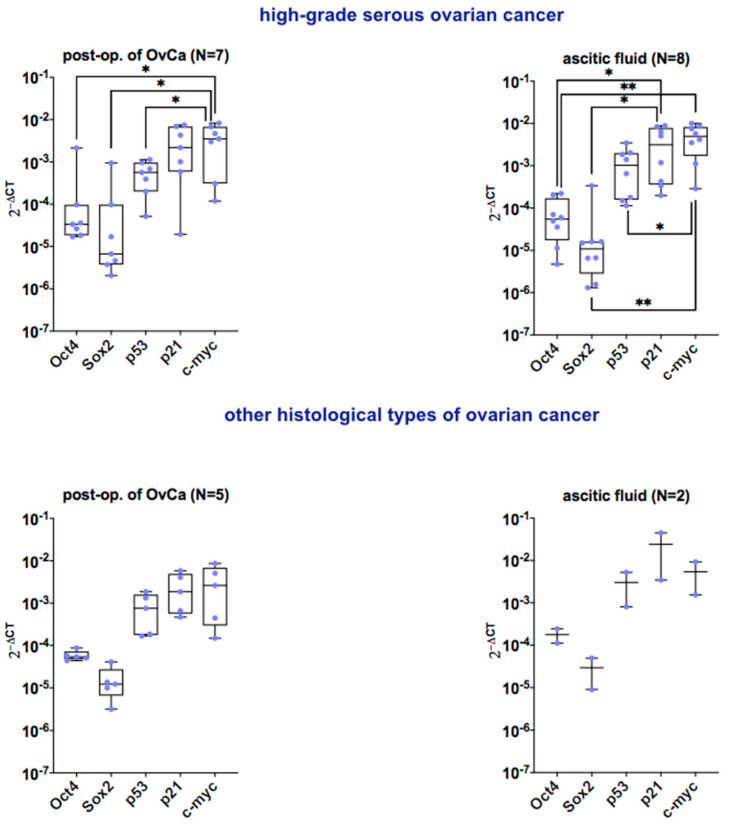
Characteristics of primary OvCa cells from post-op. of OvCa and ascitic fluid. Box and whiskers plots show 2^−ΔCT^ of selected mRNA transcripts in cells from all patients. Blue dots represent cells from each analyzed patient. Box and whiskers plots show minimal and maximal values for each transcript. Lines represent the median. The upper panel shows plots for cells from high-grade serous OvCa, and the lower panel shows plots for cells from other histological types of OvCa. ** *p* < 0.01 and * *p* < 0.05 calculated vs. each marker by one-way ANOVA. For other histological types of ovarian cancer, there were not significant differences between markers. Abbreviations: post-op. of OvCa-post-operative tissue of ovarian cancer.

**Figure 3 ijms-24-15862-f003:**
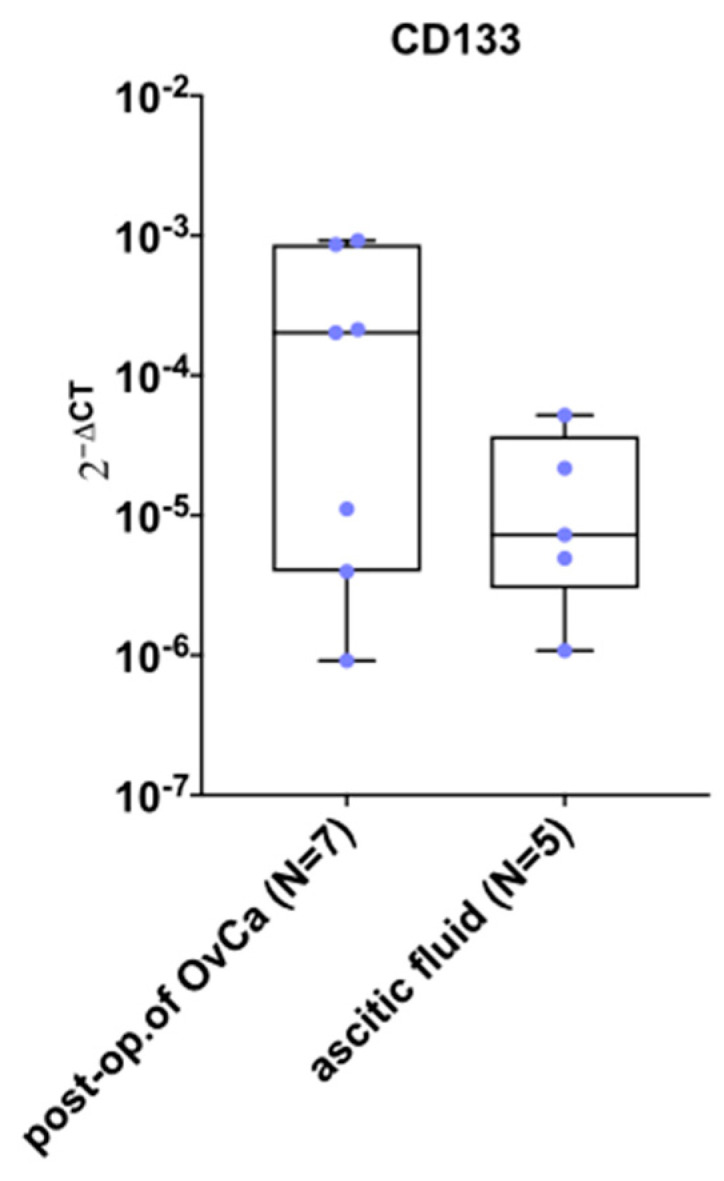
Characteristics of primary OvCa cells from post-op. tissues of OvCa and ascitic fluid. Box and whiskers plot shows expression of CD133 marker by cells from all patients. Blue dots represent cells from each analyzed patient. Whiskers show minimal and maximal values for CD133 marker. Lines represent the median. Abbreviations: post-op. of OvCa-post-operative tissue of ovarian cancer.

**Figure 4 ijms-24-15862-f004:**
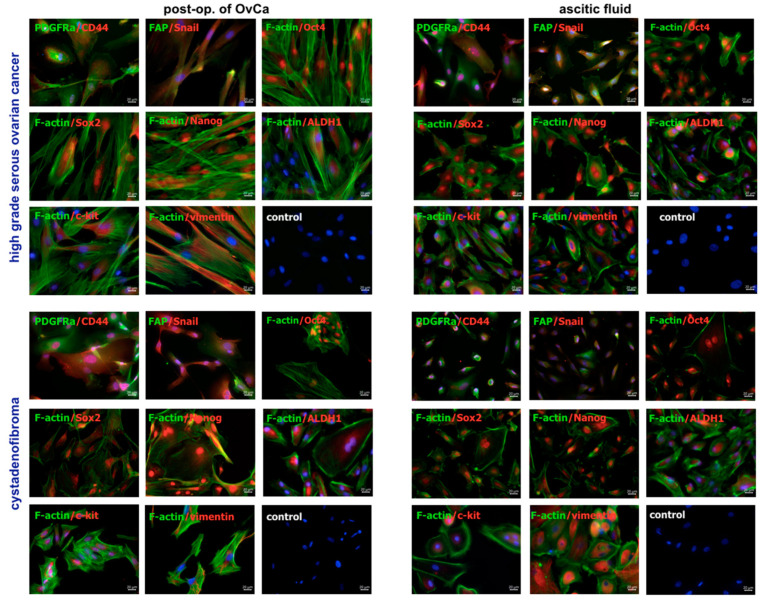
Characteristics of primary OvCa cells from post-op. of OvCa and ascitic fluid. The top panel shows cells from high-grade serous OvCa, bottom panel shows cells from cystadenofibroma. Fluorescence microscopy images: cell nuclei stained with DAPI (blue), selected markers stained with Alexa Fluor 488 (green) or Alexa Fluor 647 (red), F-actin stained with Alexa Fluor 488 phalloidin (green), bars represent 20 µm. Abbreviations: post-op. of OvCa—post-operative tissue of ovarian cancer.

**Figure 5 ijms-24-15862-f005:**
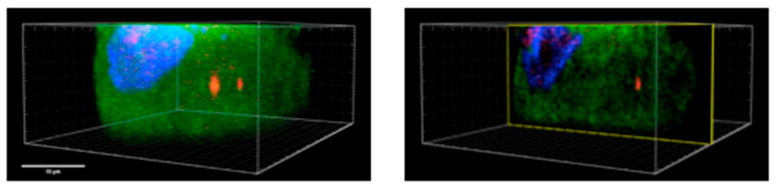
Confocal microscopy imaging of HATMSC2-MVs internalization into primary OvCa cells. Cells treated with HATMSC2-MVs at the ratio 100:1 for 24 h (cell nuclei stained with DAPI in blue, cytoplasmic expression of PDGFRα stained with Alexa Fluor 488 in green, HATMSC2-MVs stained with DiD in red). Bar represents 10 µm. HATMSC2-MVs—microvesicles derived from immortalized human mesenchymal stem cells of adipose tissue origin.

**Figure 6 ijms-24-15862-f006:**
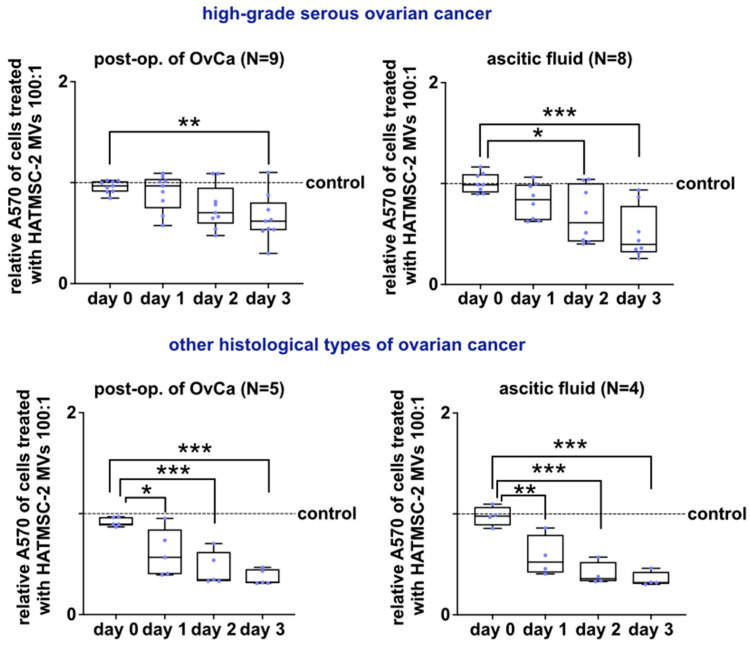
Effect of HATMSC2-MVs on the metabolic activity of primary OvCa cells from post-op. tissue and ascitic fluid. Metabolic activity of primary OvCa cells cultured in standard conditions was assessed using an MTT assay on days 0, 1, 2, and 3 following treatment with HATMSC2-MVs at a ratio of 100:1. Untreated cells without MVs served as a control. Box and whiskers plots show the relative absorbance of cells from all patients treated with HATMSC2-MVs. Blue dots represent cells from each analyzed patient. Whiskers show minimal and maximal values of analyzed cells. Lines represent the median. The top panel shows plots for cells from high-grade serous OvCa, and the bottom panel shows plots for cells from other histological types of OvCa. *** *p* < 0.001, ** *p* < 0.01, * *p* < 0.05 calculated vs. day 0 by one-way ANOVA. Abbreviations: post-op. of OvCa—post-operative tissue of ovarian cancer; HATMSC2-MVs—microvesicles derived from immortalized human mesenchymal stem cells of adipose tissue origin.

**Figure 7 ijms-24-15862-f007:**
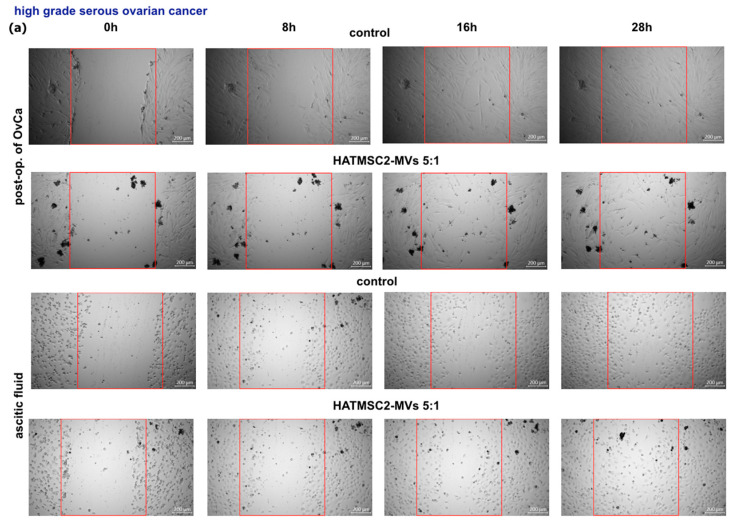
Effect of HATMSC2-MVs on the migration activity of primary OvCa cells from post-op tissues and ascitic fluid. Migration activity of primary OvCa cells cultured in standard conditions was measured using a scratch test following treatment with HATMSC2-MVs at a ratio of 5:1. Untreated cells without MVs served as a control. (**a**) Representative images of scratch closure (measured between the red lines) by cells from patients of high-grade serous OvCa on 0 h, 8 h, 16 h, and 28 h. The top panel shows control and HATMSC2-MVs treated cells from post-op. tumor of OvCa, and the bottom panel shows cells from ascitic fluid. Bars represent 200 µm. (**b**) Representative images of cells from patients with cystadenofibroma on 0 h, 8 h, 16 h, and 28 h. The top panel shows control and HATMSC2-MVs-treated cells from post-operative tumor of OvCa, and the bottom panel shows cells from ascitic fluid. Bars represent 200 µm. (**c**) Box and whiskers plots show relative scratch closure for cells from all samples treated with HATMSC2-MVs. Blue dots represent cells from each analyzed patient. Whiskers show minimal and maximal values for each cell type. Lines represent the median. The left plot shows values for cells from high-grade serous OvCa, and the right plot shows values for cells from other histological types of OvCa. Abbreviations: post-op. of OvCa—post-operative tissue of ovarian cancer. HATMSC2-MVs—microvesicles derived from immortalized human mesenchymal stem cells of adipose tissue origin.

**Figure 8 ijms-24-15862-f008:**
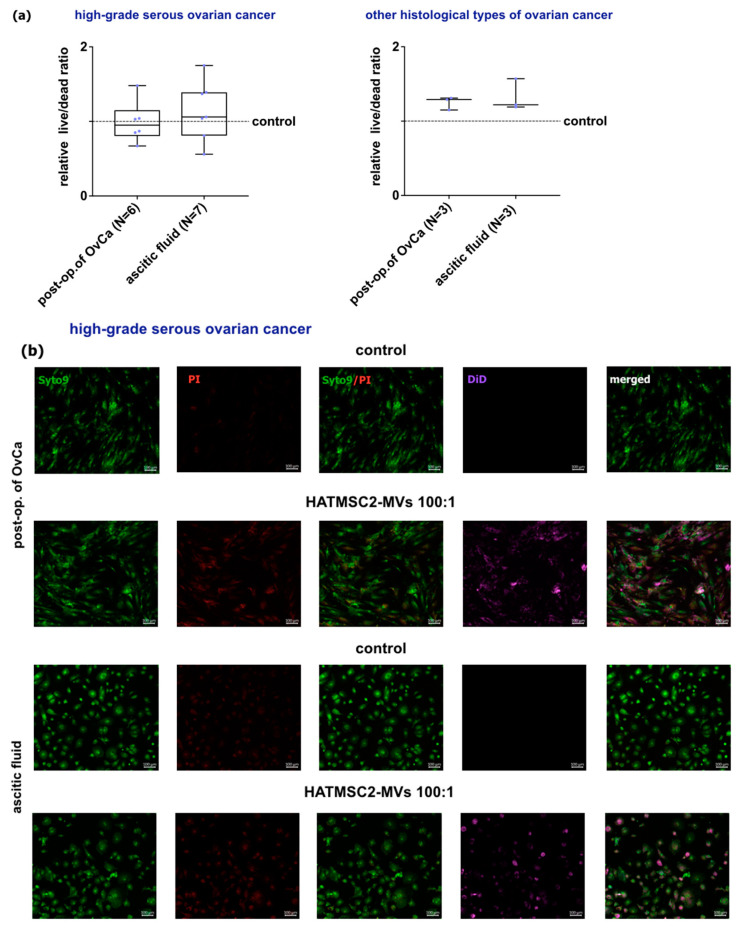
Effect of HATMSC2-MVs on the survival of primary OvCa cells from post-op tissues and ascitic fluid. Survival of cells following treatment with HATMSC2-MVs at a ratio of 100:1 was assessed by confocal microscopy imaging. Untreated cells without MVs served as a control. Live cells stained with Syto 9 (green), dead cells stained with propidium iodide (red), HAT-MSC2-MVs stained with DiD (violet). (**a**) Box and whiskers plots show relative ratio of live to dead channel. Whiskers show minimal and maximal values for each cell type. Lines represent the median. Blue dots represent cells from each analyzed patient. (**b**) Representative confocal microscopy images of cells from high-grade serous OvCa. The top panel shows control and HATMSC2-MVs-treated cells from post-op tumor of OvCa, and the bottom panel shows cells from ascitic fluid. Bars represent 100 µm. (**c**) Representative confocal microscopy images of cells from cystadenofibroma. The top panel shows control and HATMSC2-MVs-treated cells from post-operative OvCa, and the bottom panel shows cells from ascitic fluid. Bars represent 100 µm. Abbreviation: post-op. of OvCa—post-operative tissue of ovarian cancer. HATMSC2-MVs—microvesicles derived from immortalized human mesenchymal stem cells of adipose tissue origin.

**Figure 9 ijms-24-15862-f009:**
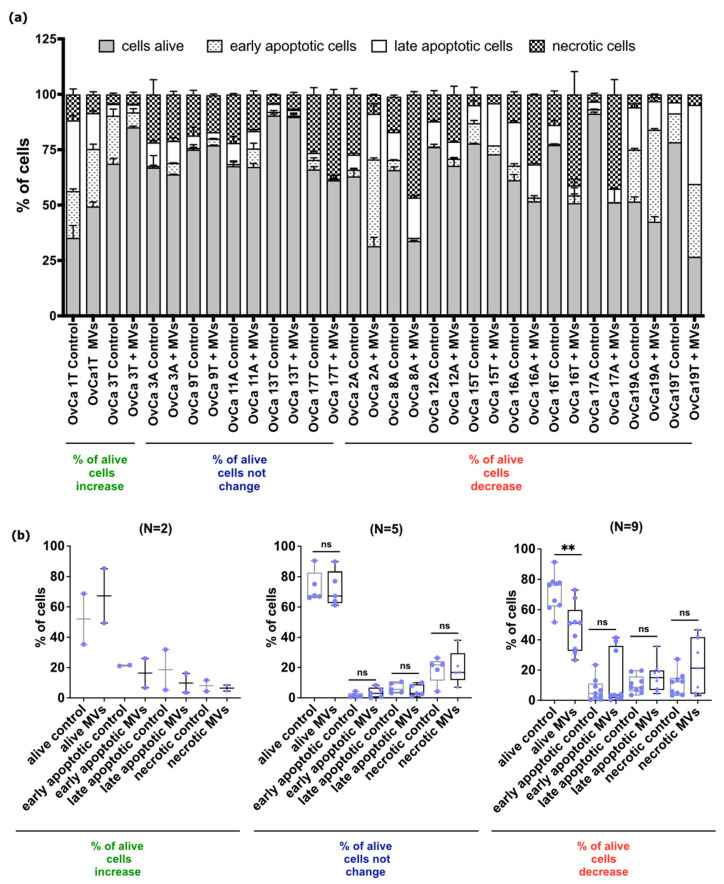
Effect of HATMSC2-MVs on the survival of primary OvCa cells isolated from post-op. tissues of OvCa and ascitic fluid. Survival of cells following treatment with HATMSC2-MVs at a ratio of 100:1 was assessed by flow cytometry. Untreated cells without MVs served as a control. (**a**) Quantification of cell viability after treatment with HATMSC2-MVs for 72 h was determined by the presence of apoptotic and necrotic cells via the double-staining of cells with propidium iodide and Annexin V. The percentages of alive, early apoptotic, late apoptotic, and necrotic cells were determined using Flowing Software 2. (**b**) Box and whiskers plots show percentage of alive cells, early apoptotic, late apoptotic, and necrotic cells for control cells and cells treated with HATMSC2-MVs. Blue dots represent cells from each analyzed patient. Whiskers show minimal and maximal values for each cell type. Lines represent the median. Untreated cells served as a control. ** *p* < 0.01 calculated vs. each respective control by Mann–Whitney test. Abbreviations: A—ascitic fluid, T—ovarian tumor tissue, MVs—microvesicles derived from immortalized human mesenchymal stem cells of adipose tissue origin, ns—not significant.

**Figure 10 ijms-24-15862-f010:**
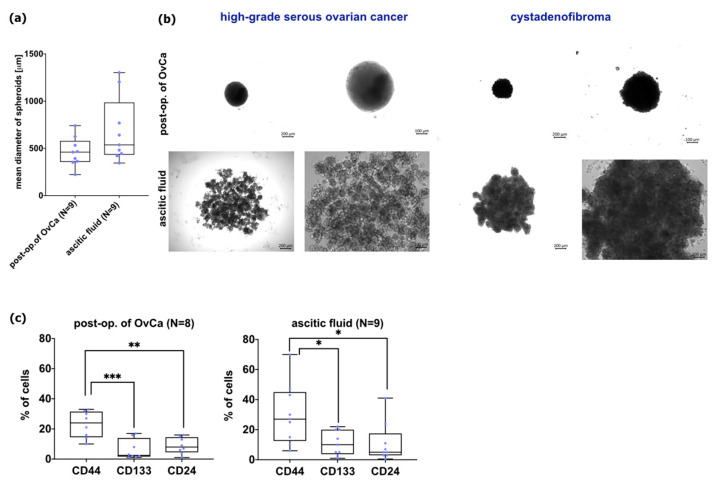
Characteristics of primary OvCa spheroids derived from post-op tissues of OvCa and ascitic fluid. (**a**) Box and whiskers plot shows the mean diameter of spheroids. Blue dots represent spheroids derived from one patient. Whiskers show minimal and maximal values for spheroid of each patient. Lines represent the median. (**b**) Representative images of spheroids. The left panel shows spheroids from high-grade serous OvCa cells, and the right panel shows representative images of spheroids cells from cystadenofibroma. Bars represent 200 µm and 100 µm. (**c**) Box and whiskers plots show the expression of selected CSCs markers on cells formed spheroids. Cells were stained with selected antibodies conjugated with fluorochromes. Blue dots represent cells from each analyzed patient. Whiskers show minimal and maximal values for each marker. Lines represent the median. *** *p* < 0.001, ** *p* < 0.01, * *p* < 0.05 calculated vs. each marker by one-way ANOVA. Abbreviations: post-op of OvCa—post-operative ovarian cancer.

**Figure 11 ijms-24-15862-f011:**
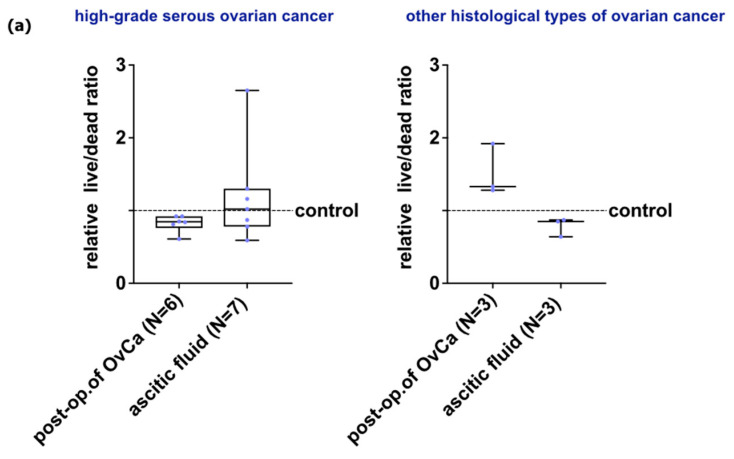
Effect of HATMSC2-MVs on the survival of primary OvCa spheroids. Effect of HATMSC2-MVs on the survival of primary OvCa spheroids derived from post-op. tumor and ascitic fluid. Survival of cells test following treatment with HATMSC2-MVs at a ratio of 100:1 was assessed by confocal microscopy imaging. Untreated cells without MVs served as a control. (**a**) Box and whiskers plots show relative ratio of live to dead channel. Blue dots represent cells from each analyzed patient. Whiskers show minimal and maximal values for each spheroid. Lines represent the median. (**b**) Representative confocal microscopy images of spheroids from high-grade serous OvCa and from cystadenofibroma. Live cells stained with Syto 9 (green), dead cells stained with propidium iodide (red), HATMSC2-MVs stained with DiD (violet). Bars represent 100 µm. The top panel shows control and treated spheroids from high-grade serous OvCa and the bottom panel shows spheroids from cystadenofibroma. Abbreviations: post-op. of OvCa-post-operative ovarian cancer; HATMSC2-MVs—microvesicles derived from immortalized human mesenchymal stem cells of adipose tissue origin.

**Figure 12 ijms-24-15862-f012:**
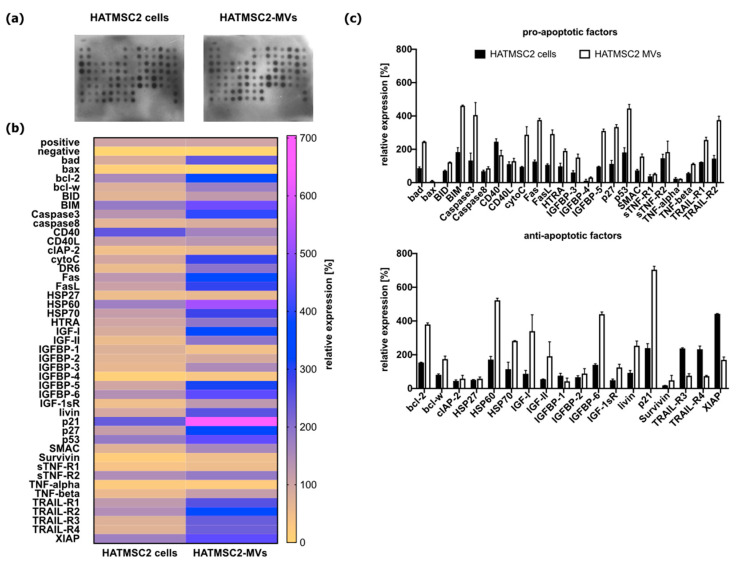
Composition of bioactive factors of the HATMSC2 cells and the HATMSC2—MVs. (**a**) Scans of representative protein arrays. The signal intensity for each antibody spot is proportional to the relative concentration of the protein in the sample. (**b**) Heat map of protein levels for the HATMSC2 cells and the HATMSC2-MVs; the magenta and yellow indicate higher and lower expression limits, respectively. Outstanding values (above 100% of positive control) are depicted in blue and magenta. Data are normalized to the internal positive control spots, which are consistent between the arrays and represent 100%. (**c**) Column graph representing bioactive factors and molecules regulating apoptosis. The data are presented as mean ± SEM values from a duplicate assessment. Abbreviations: HATMSC2-MVs—microvesicles derived from immortalized human mesenchymal stem cells of adipose tissue origin.

**Table 1 ijms-24-15862-t001:** Characteristics of patients.

No.	Patient ID	Patient’s Age	OvCa Materials Collection		Histological Type
			Post-Op. Tumor (T)	Ascitic Fluid (A)	
1	OvCa1	56	1T	NA	Ovarian clear cell carcinoma
2	OvCa2	60	NA	2A	Ovarian clear cell carcinoma
3	OvCa3	70	3T	3A	High-grade serous ovarian cancer
4	OvCa 6	66	6T	NA	High-grade serous ovarian cancer
5	OvCa 7	54	NA	7A	High-grade serous ovarian cancer
6	OvCa 8	65	8T	8A	Cystadenofibroma
7	OvCa 9	82	9T	9T	High-grade serous ovarian cancer
8	OvCa10	59	10T	NA	Mucinous ovarian cancer
9	OvCa11	53	11T	11A	Mucinous ovarian cancer
10	OvCa12	52	12T	12A	High-grade serous ovarian cancer
11	OvCa13	75	13T	13A	High-grade serous ovarian cancer
12	OvCa14	69	14T	NA	Malignant mixed Mullerian tumor called carcinosarcoma
13	OvCa15	54	15T	15A	High-grade serous ovarian cancer
14	OvCa16	72	16T	16A	High-grade serous ovarian cancer
15	OvCa17	64	17A	17T	High-grade serous ovarian cancer
16	OvCa19	70	19A	19T	High-grade serous ovarian cancer

Abbreviations: NA—not applicable.

## Data Availability

All data generated or analyzed during this study are included in this article.

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
