# Peer review of "Mesenchymal Stem Cell Microvesicles from Adipose Tissue: Unraveling Their Impact on Primary Ovarian Cancer Cells and Their Therapeutic Opportunities"

_ijms, 2023, doi:10.3390/ijms242115862_

Round 1

Reviewer 1 Report

Comments and Suggestions for Authors

Mesenchymal Stem Cell Microvesicles from Adipose Tissue: Unraveling their Impact on Primary Ovarian Cancer Cells and their Therapeutic Opportunities

The study provides some phenotypic and molecular characterization of ovarian cancer samples (OvCa) and then test whether microvesicles (MV) derived from their preestablished HATMSC2 cell line can provide therapeutic potential against OvCa of different type and grades.

The study is very well written, and both the introduction and discussion are quite thorough. The study is quite comprehensive with roughly a total of 20 primary OvCa samples included. While the data is well presented and figures of high quality. Here are some suggestions for improvements.

1.      The author spend much effort characterizing the phenotypic and molecular profiles of their primary OvCa samples (Fig 1, 2, 4 and 4). However, that fata once presented is done for. Am wondering if the author tried to correlate phenotype and/or molecular signature with the sensitivity or functional reaction of the OvCa samples to the HATMSC2- MVs? I understand that the author present the data based on the OvCa type and grade, which is great. But did the author tried to make sense of phenotype/molecular profile vs functional outcome interpedently of the type and grade of cancer? This could be indicated in the discussion or included as a limitation.

2.      The author state that CD34 (page 3, line 144) is a hematopoietic (HP) marker. While it is a marker for HP stem and progenitors, it is not only expressed by those, as it is also expressed by endothelial cells. So authors should avoid stating that its specific to HP cells, but could leave as a marker for HP stem and progenitors.

3.      Did the author characterize the size of HATMSC2- MVs in the past? If so they could indicate in Introduction.

4.      Not sure that panel A in Fig 9 is required. The data supports panel B, which presents and overview of those results. The description of those results in the text (page 11-14) is quite lengthy and am not convinced that it is really required. The main results should be presented in a concise manner and data not required to be presented in bracket for all as done, since it can be inferred in figure 9B.

5.      Authors should state the ratio meaning of HATMSC2- MVs to OvCa when first used. Currently, it is defined in line 287, but instead, it should be defined in line 265.

6.      The authors state that Mann-Whitney U test was used for determining statistical significances. However, a great of the graph presented would require one-way (or two-way for some)  ANOVA analysis (number of groups 3). For those graphs, author should run new statistical analysis to determine if differences were significant or not.

7.      Another limitation of the current study not stated is that its unclear whether the effects induced by the MVs related to the proteins content described by the protein array used (fig 12). So authors need to carefully word their conclusions accordingly, (e.g. abstract overstates that the MV induce their anti-cancer effects through the delivery of those molecules…at this time, their data does not support that) and stress the limitation and need to follow up with functional work.

Reviewer 2 Report

Comments and Suggestions for Authors

I thank the authors for an interesting study. In the study, the effect of therapy by using HATMSC2-MVs has been assessed on primary OvCa cells. The possibility to inhibit OvCa cells growth and apoptosis induction after exposure of OvCa cells on HATMSC2-MVs treatment has been demonstrated. The results obtained are important for the development of cancer therapy methods.

However, there are a few remarks that must be resolved:

1. Unfortunately, there is no description of the preparation and characteristics of HATMSC2-MVs. Please use MISEV recommendations [J Extracell Vesicles, 2018, 10.1080/20013078.2018.1535750]

2. MVs originating from tumor cells can carry on their surface the tumor receptors described in section 2.1.1. It would be interesting to compare the expression of specific receptors in tumor cells and MVs.

3. Section 2.7. Bioactive factors of HATMSC2-MVs were analyzed. However, the presence of these factors indicates insufficient purity of MVs, and these factors are co-released impurities. All of the above factors cannot be considered cargo for MVs due to their small size. You can read more in the article: [J Extracell Vesicles, 2018, 10.1080/20013078.2018.1535750] and [Bioessays, 2012, 10.1002/bies.201200045]

Therefore, a conclusion about enhanced expression of some factors in MVs cannot be made.

4. Fig. 12b. The units of measurement on the scale should be indicated.

Round 2

Reviewer 2 Report

Comments and Suggestions for Authors

The new additions to the manuscript made a big difference. The quality of the paper had improved, and all my questions were addressed. No more comments.